# Consistent Structural Relation Learning for Zero-Shot Segmentation

**Peike Li**[1,2]\* **Yunchao Wei**[1], **Yi Yang**[1]
[1]ReLER Lab, Australian Artificial Intelligence Institute
University of Technology Sydney
[2]Baidu Research
peike.li@student.uts.edu.au, {yunchao.wei, yi.yang}@uts.edu.au

## Abstract

Zero-shot semantic segmentation aims to recognize the semantics of pixels from unseen categories with zero training samples. Previous practice [1] proposed to train the classifiers for unseen categories using the visual features generated from semantic word embeddings. However, the generator is merely learned on the seen categories while no constraint is applied to the unseen categories, leading to poor generalization ability. In this work, we propose a Consistent Structural Relation Learning (CSRL) approach to constrain the generating of unseen visual features by exploiting the structural relations between seen and unseen categories. We observe that different categories are usually with similar relations in either semantic word embedding space or visual feature space. This observation motivates us to harness the similarity of category-level relations on the semantic word embedding space to learn a better visual feature generator. Concretely, by exploring the pair-wise and list-wise structures, we impose the relations of generated visual features to be consistent with their counterparts in the semantic word embedding space. In this way, the relations between seen and unseen categories will be transferred to implicitly constrain the generator to produce relation-consistent unseen visual features. We conduct extensive experiments on Pascal-VOC and Pascal-Context benchmarks. The proposed CSRL outperforms existing state-of-the-art methods by a large margin, resulting in ~7-12% on Pascal-VOC and ~2-5% on Pascal-Context.

## 1  Introduction

Semantic segmentation [2, 3] is a fundamental computer vision task that aims to assign a semantic label to each pixel in the given image. Although the development of FCN-based models [4, 5, 6] has significantly advanced semantic segmentation, the success of these approaches highly relies on cost-intensive and time-consuming dense mask annotations to train the network. To relieve the human effort in annotating accurate pixel-wise masks, there is an increasing interest in weakly-supervised segmentation and few-shot segmentation methods. Weakly supervised segmentation [7, 8] targets on learning segmentation models using lower-quality annotations such as image-level labels [9, 10], bounding boxes [11, 12] and scribbles [13, 14], which can be obtained more efficiently compared to pixel-wise masks. Meanwhile, few-shot segmentation [15, 16, 17, 18, 19] tackles the semantic segmentation from a meta-learning perspective and aims to perform segmentation with only a few annotated samples. Even significant progress has been made, these works are hard to completely liberate the request for mask annotations.

Most recently, Bucher *et al.* [1] took a step further to investigate how to effortlessly recognize those never-seen categories with zero training examples, and proposed a new learning paradigm,

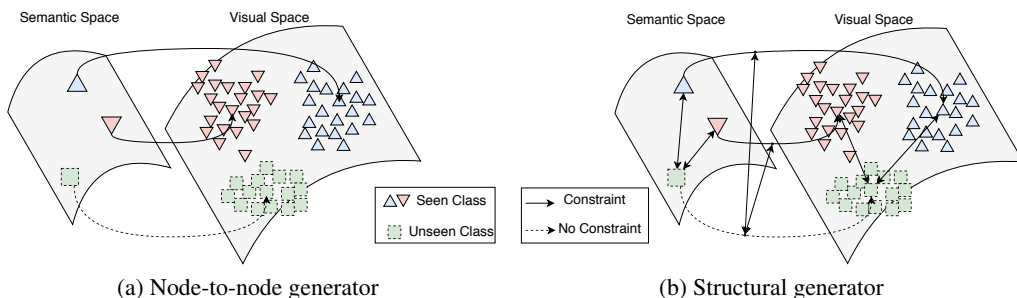

| (a) Node-to-node generator | (b) Structural generator |

Figure 1: **Illustration of CSRL**. To achieve the goal of GZS3, we learn a generator to produce visual features from semantic word embeddings. Compared to (a) node-to-node generator, the proposed (b) structural generator explores the structural relations between seen and unseen categories to constrain the generation of unseen visual features.

named Generalized Zero-Shot Semantic Segmentation (GZS3). Specifically, during the training phase, in addition to the annotated images of seen categories, we are also provided with the semantic word embeddings of both seen and unseen labels. At test time, GZS3 aims to segment images containing pixels of all categories. As zero training examples of unseen categories are available, the key challenge of GZS3 lies in how to correctly recognize the pixels from these unseen categories. To tackle this, Bucher *et al*. [1] proposed a generative method by exploiting semantic word embeddings to generate unseen visual features, which are further employed to learn the classifiers for conducting segmentation. However, when training the generator from semantic space to visual space, they take each category independently with merely node-to-node knowledge transfer of seen categories. As shown in Figure 1a, no constraint is applied to guarantee the quality of generated visual features of unseen categories, resulting in poor generalization ability.

Hence, we seek to harness the inter-class relationship between seen and unseen categories to learn a better generator. We observe that different categories are roughly with similar relations in either semantic word embedding space or visual feature space. Therefore, we assume the relational structure embedded in the semantic space can be conveniently transferred to constrain the generated visual features of unseen categories. To this end, we propose Consistent Structural Relation Learning (CSRL) framework to tackle the challenging GZS3 task. Particularly, we propose a semantic-visual structural generator by integrating both feature generating and relation learning in a unified network architecture. Instead of taking each category independently, our CSRL generates the visual features from both seen and unseen categories, simultaneously. We additionally introduce the relational constraints from different structure granularities, including point-wise, pair-wise, and list-wise consistency, to facilitate the generalization of unseen categories. In this way, the learned visual features will be imposed to keep a consistent relational structure to their semantic-based counterparts, making the generator better adapt to unseen categories. Following [1], we conduct extensive experiments on two GZS3 benchmarks based on Pascal-VOC and Pascal-Context datasets. The proposed CSRL outperforms existing state-of-the-art methods by a large margin, resulting in ~7-12% on Pascal-VOC and ~2-5% on Pascal-Context.

## 2 Related Work

**Zero-Shot Learning** ZSL [20, 21] aims to recognize unseen classes with no training examples by leveraging the semantic label embeddings (*e.g.*, word embeddings or attribute vectors) as side information. Despite on the traditional image classification task, ZSL has been applied to predict novel action in videos [22, 23], detect unseen objects [24, 25], and recently, to segment pixel-wise unseen categories [26, 1]. Former practices address ZSL by learning a projection function from visual space to semantic space [27, 28] or model weight space [29]. However, the intra-class variation in visual space is neglected by mapping to a deterministic word embedding in semantic space. Recently, due to the advance of deep generative models [30, 31], one can overcome the scarce of unseen visual features by directly generating samples from semantic word embeddings. Commonly, these generative-based methods [32, 33] train their models firstly on seen classes and then generate visual features for unseen classes. However, the quality of the generated unseen features solely relies on

the generalization ability of the generator. Differently, in this work, we apply structural relation consistency as constraints to guide the learning process.

**Generalized Zero-shot Semantic Segmentation**   Semantic segmentation under fully supervised paradigm [34, 35, 36, 37, 38, 39] and domain adaptation scheme [40, 41, 42] are extensively studied. To extremely reduce the cost of label annotation, previous works focus on weakly-supervised segmentation [7, 8, 43] and few-shot segmentation [17, 44]. Most recent works [1, 26] further extend the zero-shot learning to the semantic segmentation task. The semantic word embeddings are projected to synthetic visual features [1] and classifier weights [26]. However, the structural relations between seen and unseen classes are not well explored. In this work, instead of simple node-to-node mapping, we tackle the zero-shot segmentation from a new perspective as structural relation learning from semantic space to visual space.

## 3  Preliminaries

We denote a set of seen classes as $\mathcal{S}$ and a disjoint set of unseen classes as $\mathcal{U}$, where $\mathcal{S} \cap \mathcal{U} = \emptyset$. Let $\mathcal{D}_s = \{(\mathbf{x}, y | \mathbf{x} \in \mathcal{X}, y \in \mathcal{Y}^s\}$ represents the set of labeled training data on seen classes, where $\mathbf{x}$ is the pixel-wise feature embeddings from the visual space $\mathcal{X} \in \mathbb{R}^{d_v}$, $y$ is the corresponding label in the label space $\mathcal{Y}^s$ of seen classes. Similar to the generalized zero-shot learning setting, in the task of GZS3, we aim to learn a model that takes an image as input and predicts the label of each pixel among both seen and unseen classes $\mathcal{S} \cup \mathcal{U}$. Clearly, without any side information, zero-shot learning is infeasible as there are no training samples of unseen classes. Thus, to achieve the goal of zero-shot learning, except the training set $\mathcal{D}_s$, we are also provided with the semantic word embeddings $\{\mathbf{a}_j | \mathbf{a}_j \in \mathcal{A}\}_{j=1}^{|\mathcal{S} \cup \mathcal{U}|}$ for both seen and unseen classes, where the semantic space $\mathcal{A} \in \mathbb{R}^{d_w}$. The $d_w$-dimensional semantic embeddings could be word representations (*e.g.*, word2vec [45] or GloVe embeddings [46]) or class attribute vectors [47]. In order to overcome the absence of unseen visual features, recent works [32, 33] adopt the generative model to produce unseen visual features. Specially, a generator $\mathcal{G} : \mathcal{A} \rightarrow \mathcal{X}$ is learned to generate visual features using corresponding word embeddings as input. Another benefit of these generative-based methods is that one can achieve the goal of zero-shot learning by directly adopting the existing CNN model (*e.g.*, Deeplab) without complex architecture modification. Concretely, the generator $\mathcal{G}$ is learned on seen classes and then generate visual features for unseen classes. A new classifier (usually the last layer of CNN) is retrained on real seen visual features and generated unseen visual features. At test time, the label of each pixel is predicted by selecting the category with the largest probability.

## 4  Methodology

As shown in Figure 2, we illustrate the details of the proposed CSRL framework. The goal of CSRL is to learn a better generator to produce visual features using semantic word embeddings as input. To achieve this goal, we introduce a semantic-visual structural generator to alternately update the node features of each category and the inter-category relations. We further exploit the structural relation consistency between seen and unseen categories to constrain the generating of unseen visual features. These structural relations include the point-wise, pair-wise and list-wise relations between seen and unseen categories. The generalized zero-shot semantic segmentation is achieved by learning on real seen visual features and the generated unseen visual features.

### 4.1  Semantic-Visual Structural Generator

Given a set of semantic word embeddings including samples from both seen and unseen categories, we aim to generate the corresponding set of synthetic visual features considering the relationships among categories. Such semantic-to-visual generation is achieved by a node-edge graph $\mathcal{G} = (\mathcal{V}, \mathcal{E})$, called semantic-visual structural generator in this work. The nodes $\mathcal{V} := \{\mathbf{v}_{i,n} | \forall i \in [1, |\mathcal{S} \cup \mathcal{U}|], n \in [1, N]\}$ in the graph denote the pixel-level feature embeddings with total $N$ samples for category $i$. The edges $\mathcal{E} := \{e_{ij} | \forall i, j \in [1, |\mathcal{S} \cup \mathcal{U}|]\}$ are constructed based on the relationships between prototypes of category $i$ and $j$.

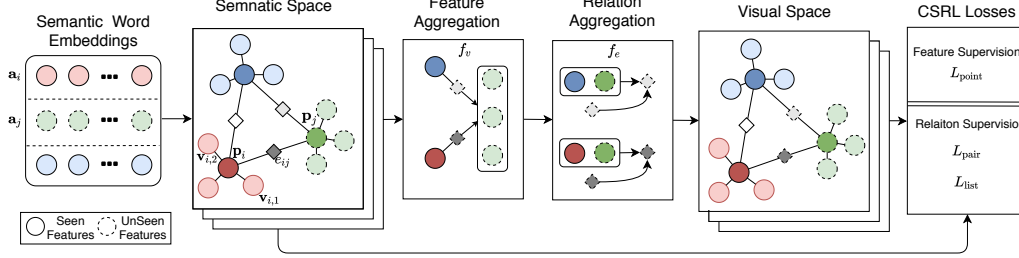

Figure 2: **The framework of the proposed CSRL.** Our CSRL incorporates the feature generating and relation learning into a unified architecture. Given the semantic word embedding, CSRL generates visual features by alternately feature and relation aggregation. The proposed CSRL is trained under supervision from point-wise consistency on seen classes, pair-wise and list-wise consistency across seen and unseen classes.

The structural generator consists of $L$ layers, where each layer contains a feature aggregation step to update the node feature and a relation aggregation step to update the edge feature. We denote $\mathbf{v}_i^\ell$ and $e_{ij}^\ell$ as the node feature and the edge feature of layer $\ell \in [1, L]$, respectively.

As the semantic word embedding $\mathbf{a}_i$ is a deterministic value, we enhance the feature diversity by concatenating a random variable $\mathbf{z}$ with a Gaussian distribution. Thus, node features are initialized by the semantic word embeddings $\mathbf{v}_{i,n}^0 = [\mathbf{a}_i \oplus \mathbf{z}_{i,n}]$, where $\oplus$ denotes the concatenation operation. Edge features $e_{ij}^0 = \mathbf{a_i} \cdot \mathbf{a_j}/\|\mathbf{a_i}\|_2\|\mathbf{a_j}\|_2$ are initialized by the cosine similarity between semantic word embeddings.

**Feature Aggregation**    To alleviate the issue introduced by abnormal samples, especially only a limited number of samples in one categories, we aggregate the feature representation based on the category prototypes instead of raw samples. Specially, the category prototype $\mathbf{p}_i$ is defined as,

$$\mathbf{p}_i^{\ell-1} = \frac{1}{N} \sum_{n=1}^{N} \mathbf{v}_{i,n}^{\ell-1}. \tag{1}$$

After calculating all prototype representations $\{\mathbf{p}_i | \forall i \in [1, |\mathcal{S} \cup \mathcal{U}|]\}$, we are able to propagate the relevant knowledge from other categories based on the edge features. The node feature aggregation of the $l$-th layer follows,

$$\mathbf{v}_{i,n}^\ell = f_v^\ell([\mathbf{v}_{i,n}^{\ell-1} \oplus \sum_{j=1, j\neq i}^{|\mathcal{S} \cup \mathcal{U}|} e_{ij}^{\ell-1} \mathbf{p}_i^{\ell-1}]; \phi_v^\ell). \tag{2}$$

where $f_v$ is a transformation network with parameters $\phi_v^\ell$.

**Relation Aggregation**    After aggregate the node features, the edge feature aggregation is processed based on the newly updated node features, The edge feature aggregation of the $l$-th layer follows,

$$e_{ij}^\ell = f_e^\ell(|\mathbf{p}_i^\ell - \mathbf{p}_j^\ell|; \phi_e^\ell)e_{ij}^{\ell-1}, \tag{3}$$

where $f_e$ is a transformation network with parameters $\phi_e^\ell$.

By alternately feature aggregation and the relation aggregation steps, we simultaneous achieve the feature generating and relation learning. At $\ell = L$, the output nodes are the generated visual features $\hat{\mathbf{x}}$ including both seen and unseen categories, while the edge features are the learned relations between categories.

## 4.2   Consistent Structural Relation Learning

The key to generalized zero-shot segmentation is the ability to generate visual features $\hat{\mathbf{x}} \in \hat{\mathcal{X}}$ conditioned on the semantic word embedding $\mathbf{a}$, even without access to any image pixels of this category. In order to learn a better generator, we explore the relation constraints from different structure granularities as supervision signals to train the generator $\mathcal{G}$.

**Point-wise consistency**  At training time, only the real visual features from seen categories are available to access. Thus, on these seen categories, we optimize the distribution divergence between real visual features and generated visual features as supervision signals. As this divergence reflects the consistency of every single category between real and generated visual feature distributions, here we note it as *point-wise consistency*. Here, we minimize distribution divergence on seen categories by optimizing the *maximum mean discrepancy* as,

$$\mathcal{L}_{\text{point}} = \frac{1}{|\mathcal{S}|} \sum_{c=1}^{|\mathcal{S}|} [\mathbb{E}_{\mathbf{x}, \mathbf{x}' \sim \mathcal{X}^c} K(\mathbf{x}, \mathbf{x}') + \mathbb{E}_{\hat{\mathbf{x}}, \hat{\mathbf{x}}' \sim \hat{\mathcal{X}}^c} K(\hat{\mathbf{x}}, \hat{\mathbf{x}}') - 2\mathbb{E}_{\mathbf{x} \sim \mathcal{X}^c, \hat{\mathbf{x}} \sim \hat{\mathcal{X}}^c} K(\mathbf{x}, \hat{\mathbf{x}})], \quad (4)$$

where $K$ is the Gaussian kernel with bandwidth parameter $\sigma$ defined as $K(\mathbf{x}, \mathbf{x}') = \exp(-\frac{1}{2\sigma^2} \|\mathbf{x} - \mathbf{x}'\|^2)$.

By optimizing the point-wise consistency on seen categories, there is no explicit constraint on the generation of unseen categories. Thus the quality of produced unseen features purely relies on the generalization ability of the generator. To enhance and constrain the visual feature generation especially on unseen categories, we transfer the structural relations on semantic word embedding space to the generator visual features space. In this paper, we consider the *pair-wise consistency* and *list-wise consistency*. The pair-wise relations reflect that the feature similarity between two categories, *i.e.*, one seen category and one unseen one, should be consistent on both semantic space and visual space. The list-wise relations require that the relation ranking permutation order should also be consistent on semantic space and visual space.

**Pair-wise consistency**  We extract the relation matrix between unseen and seen categories from the edge features in structural generator $\mathcal{G}$ as $\mathbf{M} = \{e_{ij}^\ell | \forall i \in [1, |\mathcal{U}|], j \in [1, |\mathcal{S}|]\} \in \mathbb{R}^{|\mathcal{U}| \times |\mathcal{S}|}$. For each unseen category, the relation values is further normalized by applying softmax function as follows,

$$\tilde{e}_{ij}^\ell = \frac{\exp(e_{ij}/\gamma)}{\sum_{j'=1}^{|\mathcal{S}|} \exp(e_{ij'}/\gamma)}, \quad (5)$$

where $\gamma$ is a scaling factor to soften the relation distribution. Thus, in the semantic word embedding space (*i.e.*, the input layer $\ell = 0$), we have the relation matrix as $\mathbf{M}^\mathcal{A}$. In the generated visual feature space (*i.e.*, the output layer $\ell = L$), the relation matrix is denote as $\mathbf{M}^{\hat{\mathcal{X}}}$.

To maintain the pair-wise relation consistency between semantic space and visual feature space, we adopt the Kullback-Leibler divergence as the learning objective. Concretely, the *pair-wise consistency* is defined as,

$$\mathcal{L}_{\text{pair}}(\mathbf{M}^\mathcal{A}, \mathbf{M}^{\hat{\mathcal{X}}}) = \frac{1}{|\mathcal{U}|} \sum_{i=1}^{|\mathcal{U}|} D_{\text{KL}}[\mathbf{M}_i^\mathcal{A} \| \mathbf{M}_i^{\hat{\mathcal{X}}}]. \quad (6)$$

**List-wise consistency**  Instead of only focus on the relationship from a pair of categories at a time, inspired by [48, 49], we further investigate the entire ranking permutation of the relation list as complementary supervision. The core idea is that we take the relation ranking as a distribution rather than a deterministic order. We aim to associate the probability with every rank permutation between semantic space and visual space. Given one permutation $\pi$ of the relation list, where $\pi(i)$ denotes the $i$-th list index of this permeation. We calculate the probability of this ranking permutation as,

$$P(\pi | \mathbf{M}_i) = \prod_{j=1}^{|\mathcal{S}|} \frac{\exp(e_{i\pi(j)}/\gamma)}{\sum_{k=j}^{|\mathcal{S}|} \exp(e_{i\pi(k)}/\gamma)} \quad (7)$$

where $\gamma$ is a scaling factor.

We aim to maintain all possible relation ranking permutations $\pi \in \mathcal{P}$ as consistent as possible both on semantic space and visual features space. Similar to pair-wise consistency, the *list-wise consistency* is defined as,

$$\mathcal{L}_{\text{list}}(\mathbf{M}^\mathcal{A}, \mathbf{M}^{\hat{\mathcal{X}}}) = \frac{1}{|\mathcal{U}|} \sum_{i=1}^{|\mathcal{U}|} D_{\text{KL}}[P(\pi \in \mathcal{P} | \mathbf{M}_i^\mathcal{A}) \| P(\pi \in \mathcal{P} | \mathbf{M}_i^{\hat{\mathcal{X}}})] \quad (8)$$

Table 1: Generalized zero-shot semantic segmentation performance on Pascal-VOC dataset.

| Settings | Methods | Seen mIoU | Unseen mIoU | Overall mIoU | Overall hIoU |
|---|---|---|---|---|---|
| unseen-2 | SegDevis | 68.1% | 3.2% | 44.1% | 6.1% |
| | SPNet | 71.8% | 34.7% | 68.2% | 46.8% |
| | ZS3Net | 72.0% | 35.4% | 68.5% | 47.5% |
| | **CSRL** | **73.4%** | **45.7%** | **70.7%** | **56.3%** |
| unseen-4 | SegDevis | 64.3% | 2.9% | 38.9% | 5.5% |
| | SPNet | 67.3% | 21.8% | 58.6% | 32.9% |
| | ZS3Net | 66.4% | 23.2% | 58.2% | 34.4% |
| | **CSRL** | **69.8%** | **31.7%** | **62.5%** | **43.6%** |
| unseen-6 | SegDevis | 39.8% | 2.7% | 33.4% | 5.1% |
| | SPNet | 64.5% | 20.1% | 51.8% | 30.6% |
| | ZS3Net | 47.3% | 24.2% | 40.7% | 32.0% |
| | **CSRL** | **66.2%** | **29.4%** | **55.6%** | **40.7%** |
| unseen-8 | SegDevis | 35.7% | 2.0% | 24.3% | 3.8% |
| | SPNet | 61.2% | 19.9% | 45.5% | 30.0% |
| | ZS3Net | 29.2% | 22.9% | 26.8% | 25.7% |
| | **CSRL** | **62.4%** | **26.9%** | **48.8%** | **37.6%** |
| unseen-10 | SegDevis | 31.7% | 1.9% | 16.9% | 3.6% |
| | SPNet | 59.0% | 18.1% | 39.5% | 27.7% |
| | ZS3Net | 33.9% | 18.1% | 26.3% | 23.6% |
| | **CSRL** | **59.2%** | **21.0%** | **50.0%** | **31.0%** |

## 4.3 Training and Inference

In this subsection, we introduce the whole procedures to achieve GZS3. During the training stage, we start from training an off-the-shelf segmentation model (*e.g.*, DeepLabv3+) on all annotated data from seen categories. After training on seen categories, we remove the last classification layer and the remaining network serves as a visual features extractor to get the training set of seen categories $\mathcal{D}_s$. Then, we train our semantic-visual structural generator $\mathcal{G}$ under the supervision of consistent structural relation learning losses,

$$\mathcal{L}(\phi) = \mathcal{L}_{\text{point}} + \mathcal{L}_{\text{pair}} + \mathcal{L}_{\text{list}}. \tag{9}$$

To maintain simplicity, here we directly add these three terms. Once the generator $\mathcal{G}$ is trained, arbitrarily many visual features can be generated from semantic word embeddings, especially for unseen categories. In this way, we build a generated unseen training set denote as $\hat{\mathcal{D}}_u = \{\hat{\mathbf{x}}, y | \hat{\mathbf{x}} \in \hat{\mathcal{X}}, y \in \mathcal{Y}^u\}$. A new pixel-level classifier is trained on the combined training set including real seen visual features from $\mathcal{D}_s$ and generated unseen visual features from $\hat{\mathcal{D}}_u$. In this way, the new model can be used to conduct generalized zero-shot semantic segmentation of a given image that exhibit categories from both seen and unseen classes.

## 5 Experiments

### 5.1 Experiment Settings

**Datasets** We conduct experiments on two datasets including Pascal-VOC [50] and Pascal-Context [51]. Pascal-VOC focuses on object semantic segmentation scenario, which contains 10,582 training and 1,449 validation images from 20 classes. Pascal-Context targets on the scene parsing scenario, which comprises 4,998 training and 5,105 validation images from 59 classes. Following [1], we construct zero-shot segmentation setups with different number of unseen classes, including 2, 4, 6, 8 and 10 unseen classes, and all the rest ones are the seen classes. Concretely, the unseen class set is extended in an incremental manner, *i.e.*, the 4-unseen set contains the 2-unseen set. The unseen class splits are *2-cow/motorbike, 4-airplane/sofa, 6-cat/tv, 8-train/bottle, 10-chair/potted-plant* for Pascal-VOC dataset and *2-cow/motorbike, 4-sofa/cat, 6-boat/fence, 8-bird/tvmonitor, 10-keyboard/aeroplane* for Pascal-Context dataset.

**Evaluation Metrics** In our experiments, similar to the standard semantic segmentation task, we adopt mean intersection-over-union (mIoU) as the principal metric. The generalized zero-shot

Table 2: Generalized zero-shot semantic segmentation result on Pascal-Context dataset.

| Settings | Methods | Seen mIoU | Unseen mIoU | Overall mIoU | Overall hIoU |
|---|---|---|---|---|---|
| unseen-2 | SegDevis | 35.8% | 2.7% | 33.1% | 5.0% |
| | SPNet | 38.2% | 16.7% | 37.5% | 23.2% |
| | ZS3Net | 41.6% | 21.6% | 41.0% | 28.4% |
| | **CSRL** | **41.9%** | **27.8%** | **41.4%** | **33.4%** |
| unseen-4 | SegDevis | 33.4% | 2.5% | 30.7% | 4.7% |
| | SPNet | 36.3% | 18.1% | 35.1% | 24.2% |
| | ZS3Net | 37.2% | **24.9%** | 36.4% | 29.8% |
| | **CSRL** | **39.8%** | 23.9% | **38.7%** | **29.9%** |
| unseen-6 | SegDevis | 31.9% | 2.1% | 28.8% | 3.9% |
| | SPNet | 31.9% | 19.9% | 30.7% | 24.5% |
| | ZS3Net | 32.1% | 20.7% | 30.9% | 25.2% |
| | **CSRL** | **35.5%** | **22.0%** | **34.1%** | **27.2%** |
| unseen-8 | SegDevis | 22.0% | 1.7% | 19.2% | 3.2% |
| | SPNet | 28.6% | 14.3% | 26.7% | 19.1% |
| | ZS3Net | 20.9% | 16.0% | 20.3% | 18.1% |
| | **CSRL** | **31.7%** | **18.1%** | **29.9%** | **23.0%** |
| unseen-10 | SegDevis | 17.5% | 1.3% | 14.3% | 2.4% |
| | SPNet | 27.1% | 9.8% | 24.3% | 14.4% |
| | ZS3Net | 20.8% | 12.7% | 19.4% | 15.8% |
| | **CSRL** | **29.4%** | **14.6%** | **27.0%** | **19.5%** |

semantic segmentation focuses on the overall performance including both seen and unseen categories. To avoid the performance on seen categories dominates, we also report the harmonic mean (hIoU) of seen mIoU and unseen mIoU suggested by [52],

$$hIoU = \frac{2 * mIoU_s * mIoU_u}{mIoU_s + mIoU_u}. \tag{10}$$

**Implementation Details**   We choose the DeeplabV3+ [6] with ResNet-101 [53] as our segmentation network. The ImageNet [54] covers a wide range of categories, where most unseen categories are actually included. Therefore, directly adopting the publicly ImageNet pre-trained model may break the setting of zero-shot learning. To avoid the supervision leakage from unseen classes, we employ the model provided by [1], which is solely pre-trained using seen categories. For the aggregation network $f_e$ and $f_v$ in Sec 4.1, we use the multi-layer perception network proposed by [55]. We implemented our method both by the Pytorch platform and the PaddlePaddle platform, both achieving similar performance. More details of the network structure and parameter settings can be found in our supplementary materials.

### 5.2   Comparisons with State-of-the-art Methods

We compare our proposed CSRL with SegDeViSe [56], SPNet [26], ZS3Net [1]. SegDeViSe regresses semantic word features from pixel-level visual features, which is learned by maximizing the cosine similarity between the output and the target word embeddings. SPNet encodes images in the word embedding space and uses a semantic projection layer to produce class probabilities. ZS3Net is the current state-of-the-art method, which generates unseen visual features from word embeddings to achieve zero-shot segmentation. All these methods adopt the same segmentation network, *i.e.*, DeepLabV3+, for a fair comparison. The key commonality shared by these methods is: they take each category as an independent point without considering its relations to other categories. Differently, we generate the unseen visual features by exploring the structural relations between categories.

We report the performance of generalized zero-shot semantic segmentation on Pascal-VOC dataset in Table 1 and Pascal-Contex dataset in Table 2. Results of SPNet are based on our implementation, and other results of ZS3Net and SegDeVis are directly taken from paper [1]. In these two tables, first, we observe that the generative methods (*i.e.*, ZS3Net, CSRL) significantly outperforms semantic embedding-based methods (*i.e.*, SegDeViSe, SPNet). The semantic embedding-based methods, although perform well on seen categories, achieve a large performance drop for unseen ones. By leveraging structural relation consistency to better guide the generation of unseen visual features, our CSRL provides significant gains particularly on the unseen classes (*e.g.*, +10.3% for the 2-unseen

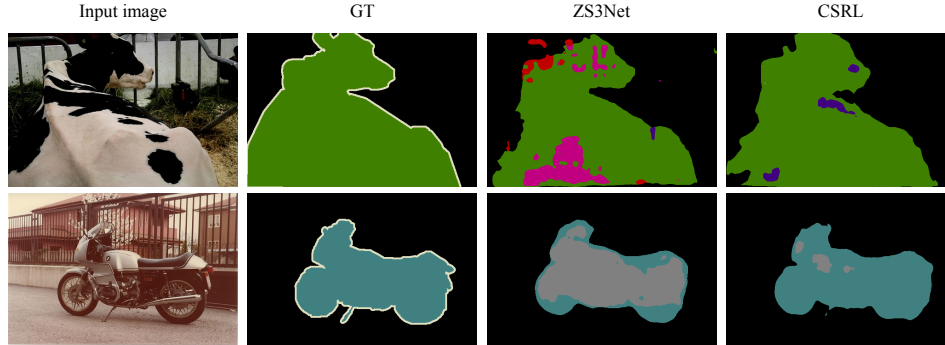

Figure 3: Qualitative comparisons on Pascal-VOC dataset under the unseen-2 setting.

split in terms of unseen mIoU). Second, our CSRL significantly outperforms others by large margins for various splits (~7-12% for hIoU), which can well demonstrate the effectiveness of the consistent structural relation learning framework. Third, our CSRL also achieves large performance gains on the more challenging benchmark Pascal-Context, which requires densely predictions for the full images. The qualitative comparison between ZS3Net and CSRL is shown in Figure 3. We can observe that our CSRL achieves much better segmentation results and successfully recognize the unseen objects (*e.g.* cow and *motorbike*) where the ZS3Net mostly fails. More qualitative results are provided in the supplementary materials.

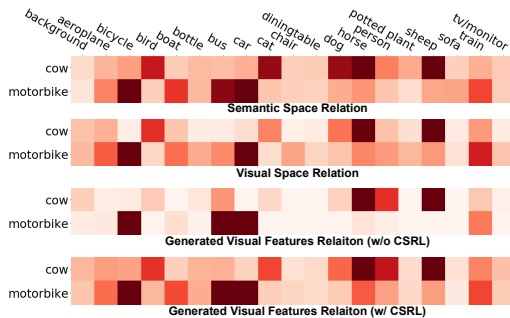

Figure 4: Relations between unseen (cow and motorbike) and seen categories.

Table 3: Ablation study of CSRL on Pascal-VOC.

| Exp | Point Pair List | | | Seen mIoU | Unseen mIoU | Overall mIoU | Overall hIoU |
|---|---|---|---|---|---|---|---|
| I | ✓ | - | - | 73.0% | 40.3% | 69.8% | 51.9% |
| II | ✓ | ✓ | - | 73.4% | 43.3% | 70.5% | 54.5% |
| III | ✓ | - | ✓ | 73.0% | 42.7% | 70.1% | 53.9% |
| CSRL | ✓ | ✓ | ✓ | **73.4%** | **45.7%** | **70.7%** | **56.3%** |

## 5.3 Ablation Analysis

**Quantitative analysis for structural relations** We conduct extensive quantitative analysis for those key components in CSRL. In Table 3, we compare the effects of different structural relations with the unseen-2 split on Pascal-VOC. First, simply performing node-to-note generation results in the hIoU score of 51.9% (I). Second, by introducing the pair-wise relation for optimization, the hIoU will be significantly enhanced by 3.6% (II). Third, by replacing pair-wise relation with list-wise relation, the improvement is still notable, *i.e.*, 2.0% (III). Finally, simultaneously considering all the components will lead to the best hIoU score of 56.3% (CSRL).

**Qualitative analysis for inter-category relations** In Figure 4, we visualize the inter-category relations between unseen categories (*i.e.*, cow and motorbike) and seen categories based on different feature embeddings. The relations are normalized for better visualization. The darker the colors, the stronger the relations. "Semantic Space Relation" and "Visual Space Relation" indicate cosine similarities of using word2vec features and CNN features with supervised training, respectively. First, we can observe that semantic relations between unseen and seen categories keep consistent across different feature spaces. Second, by introducing the CSRL, the relations of generated visual features will be more consistent compared to those without CSRL, leading to better discriminative ability.

# 6 Conclusion

In this paper, to tackle the challenging generalized zero-shot semantic segmentation task, we proposed a simple yet effective framework called Consistent Structural Relation Learning (CSRL). We propose a semantic-visual structural generator by integrating both feature generating and relation learning in a unified network architecture. We effectively explore relation consistency from multiple structure granularities to better guide the generation of unseen visual features. The proposed CSRL achieves the new state-of-the-art on two zero-shot segmentation benchmarks, which outperforming the former practices by a large margin. Although CSRL achieves a large improvement for the generalized zero-shot semantic segmentation, there is still a long way to go. We can observe that there is still a large performance gap between the seen and the unseen categories on the two benchmarks. Thus, more effective GZS3 algorithms are still required to alleviate this gap. We hope that our efforts will motivate more researchers and ease future research.

## Acknowledgment

This work is partly supported by ARC DECRA DE190101315 and ARC DP200100938.

## Broader Impact

Our research advances the zero-shot learning segmentation task, which alleviates the need for expensive human annotations when learning the unseen categories. Moreover, our research needs less computational cost which only needs to re-train the classification head rather than the whole network. Thus our research is more financially-friendly and environmental-friendly compared to the traditional fully-supervised learning paradigm. By utilizing the large amount of word embedding vectors, the network can be built with stronger scalability to potential unseen categories.

## Footnotes

\*Part of this work is done when Peike Li is an intern at Baidu Research

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
