[Supplementary Material]

# Supplementary Materials of "Structural Relation Learning for Zero-Shot Segmentation"

## A    Additional Discussion on CSRL

To better illustrate the differences of three structural consistency constraints proposed in Sec 4.2, we list the constraint scopes and potential effects of point-wise (Eq. 4), pair-wise (Eq. 6) and list-wise (Eq. 8) consistencies in Table 4. The main differences of these three losses are:

1. The point-wise consistency acts on the visual feature distribution of each seen category, while pair-wise and list-wise consistencies are applied to constrain the relations between unseen and seen categories.

2. During training, only the real visual features of seen categories are available. Thus, we minimize the feature distribution distance between real and generated visual features by *point-wise consistency*. As the real visual features of unseen categories are inaccessible, we constrain the unseen generated features by *pair-wise consistency* and *list-wise consistency*. These two terms aim to migrate the relation knowledge from the semantic word embedding space to the generated visual space.

3. The optimization goals of *pair-wise* and *list-wise* are partly coincide but have complementary advantages. The *pair-wise consistency* focuses on minimizing the pair-wise relation divergence. However, by taking the relation ranking permutation as a distribution, we explore more potential guidance information by minimizing the distribution divergence of relation ranking p.

Table 4: Comparison among different structural consistencies.

| Structural Loss | Scope | Category | Effect |
|---|---|---|---|
| point-wise | feature | seen only | minimize feature distribution distance |
| pair-wise | relation | between unseen and seen | minimize pair-wise relation distribution divergence |
| list-wise | relation | between unseen and seen | minimize relation ranking distribution divergence |

## B    More Implementation Details

Following the common practice [6, 57], the segmentation model is trained by a SGD optimizer with a polynomial learning rate decay scheduler, which has the base learning rate of $7e^{-3}$, momentum $0.9$ and weight decay $5e^{-4}$. The generative model is trained using Adam optimizer with the learning rate of $2^{-4}$. We employ the word2vec embeddings [45] with $d_w = 300$ as the semantic word embeddings. The input Gaussian noise has the same dimension as the word2vec embeddings. The visual feature dimension is $d_v = 256$. When calculating the pair-wise consistency and list-wise consistency, the softmax temperature is experimentally set as $\gamma = 0.5$. We illustrate the detailed network architectures in Figure 5. In our network, the intermediate dimension is set as 256. The slope of LeakyReLU is set as 0.2. And the dropout probability [58, 59] is 0.5. In order to save computational cost, we calculate the list-wise consistency in Eq. 8 using the permutation order with the highest probability rather than all permutation orders.

Figure 5: Detailed Network Architecture. We show the $\ell$-th layer ($\ell \in \{1, 2, 3, 4\}$) in our semantic-visual structural generator, which consists of a feature aggregation network $f_v^\ell$ and a relation aggregation network $f_e^\ell$.

## C  More Qualitative Results

Here we show more qualitative comparison results on Pascal-VOC in Figure 6 and Pascal-Context in Figure 7. We also illustrate the limitation of the proposed CSRL in Figure 8. Under complex scenarios, *e.g.*, multiple instances (row 1 in Figure 8), highly occlusion (row 2 and 4 in Figure 8) or rare scene (row 3 in Figure 8), our CSRL fails to recognize the unseen categories and leads to relatively worse segmentation results. We have to note that although CSRL achieves a large performance boost on the generalized zero-shot semantic segmentation task, due to the fact there is zero-example available during training, the performance on unseen categories is still far from satisfactory. We hope our efforts could motivate more researchers and help ease future research in zero-shot segmentation.

Figure 6: Qualitative comparisons on Pascal-VOC dataset under the unseen-2 split. The unseen categories are cow and motorbike.

Figure 7: Qualitative comparisons on Pascal-Context dataset under the unseen-2 split. The unseen categories are cow and motorbike.

Figure 8: Failure cases on Pascal-VOC (first and second row) and Pascal-Context (third and fourth row). The unseen categories are cow and motorbike, which are emphasized with the white dashed boxes.