[Reviews · NeurIPS 2020]

Review 1

Summary and Contributions: === Post rebuttal update === I originally gave this paper an '8' and I will keep my original rating. The method is a good improvement upon [1]: it extends [1] with a simple and reproducable idea. Experimentally they demonstrate good improvements over [1]. In contrast to R3, I think that is not only a decent amount of novelty, but also the simple kind of novelty that is likely to be adopted by other reviewers. The other two main weaknesses highlighted by several reviewers were: 1) A better positioning w.r.t. [1], which the authors did in their rebuttal (which should make its way to the paper). 2) Comparison to more related work, convincingly done in the rebuttal. Therefore I am satisfied with the author response and will keep my original rating. === End of post-rebuttal update === The authors address zero-shot semantic segmentation by exploiting word embeddings. In particular, [1] proposed a framework to create visual features based on (A) images of seen categories, and (B) embedding-features generated from word-embeddings, for both seen and unseen categories. They first make the seen visual features consistent with the generated seen word-embedding-features (using Maximum Mean Discrepancy). This enables generating examples in word-embedding-feature space for unseen categories (from their word-embeddings). This in turn can be used to train a classifier over both seen categories (using real visual features) and unseen categories (using word-embedding features). This paper extends [1] by: - Improving the feature generation using explicit weighted averages over nearby concepts (Sec. 4.1). - Add two 'consistency' losses which aim to preserve relations between classes in both the visual and word-embedding-space. Results show decent improvements over [1] and earlier works [25, 50].

Strengths: - Well-motivated idea. - Simple solution, which can be used and reproduced in practice. - Good results. - Well written.

Weaknesses: The following weaknesses are minor: - discussion with related work can be improved, specifically w.r.t. [1]. - The evaluation dataset (PASCAL Context) is rather old. Evaluation on a more modern dataset such as the COCO panoptic dataset or ADE20k would be desirable, especially since knowledge transfer is expected to work better when there exist more unseen classes. (Pascal Context has 33 classes, COCO has more in the range of 160, ADE20k has many more but has more label ambiguities).

Correctness: Yes.

Clarity: Yes.

Relation to Prior Work: Contribution w.r.t. [1] can be better described.

Reproducibility: Yes

Additional Feedback: The following items are suggestions to improve the paper. Addressing the first three in the author response is optional. Please don't address the others. 1. Please better discuss the differences w.r.t. [1]. In particular: how does Sec. 4.1 differ from their method? Please mark more clearly that the point-wise consistency was already used in [1]. 2. How is \gamma determined? is your system sensitive to its value? 3. It would be great if the authors could apply their method to the COCO panoptic dataset (or maybe ADE20k). This dataset seems more suitable for this task (see motivation above). Having such number would set a baseline and a new standard for future work. 4. The Reference section has et al. instead of the full authors list. This is unacceptable and needs to be changed to include the names of all authors. 5. Please discuss how the current work relates to similar work in Computer Vision: https://arxiv.org/abs/1703.08769 6 L28: "lower quality annotations". This is false. The quality may be the same or higher (since segmentation is harder to do). Please rephrase. For example 'weaker form of annotations' or 'annotations with less information than the target task'. 7. L140: At l=L [...]. It would be clearer to relate \hat(x) to v^l_{i,j} directly. 8. L157-L165: This paragraph is now incorrectly part of 'point-wise consistency'. Consider moving to the beginning of 4.2 or put its content into the correct paragraphs below (pair-wise consistency and list-wise consistency).


Review 2

Summary and Contributions: This paper proposes a novel feature generation approach for zero-shot semantic segmentation. The key idea is to constrain the generating of unseen visual features by exploiting the structural relations between seen and unseen categories. They show better performance compared to previous works.

Strengths: -The proposed approach is novel. -Results are descent.

Weaknesses: -It is not writtent clearly whether the pixels of unseen classes are used during training or not. If yes, then it is not fair to compare with the Z3SNet and SPNet. In Section 3, the problem statement describes the standard zero-shot setting without using any pixels of unseen classes. But in Section 4.1, the nodes consist of pixel feature embeddings of unseen classes, which contradicts the setup described in Section 3. -Feature generation has been extensively explored in the context of zero-shot image classification. But this paper fails to compare with other feature generation methods except [1]. -This paper looks like another feature generation approach for zero-shot image classification. There is no specific technic designed for semantic segmentation. The authors did not justify why the proposed feature generator works well for the semantic segmentation problem. --------------------------- Post-rebutall: My concerns regarding the weakness are addressed in the rebuttal. Therefore I decide to increase my rating to be an "7"

Correctness: Yes.

Clarity: The paper writing needs to be improved. The problem setup is not very clear. In particular, it is unclear if pixels of unseen classes are used, how the background pixels are handled and what happen if unseen and seen classes co-occurrence in one image.

Relation to Prior Work: Yes, the authors discuss the differences.

Reproducibility: No

Additional Feedback:


Review 3

Summary and Contributions: This paper proposes an approach for zero-shot segmentation tasks by considering relations between different categories. In particular, the framework is conditioned on semantic word embeddings and it tries to generate visual features of unseen classes by using the similiarity between seen classes and unseen classes. And such similarity is modeled by the relation aggregation and the pair-wise and list-wise consistency operations. The authors conducted experiments on Pascal-voc datasets and better performance is achieved on these two datasets.

Strengths: 1. The problem of zero-shot segmentation is valuable, since collecting labels for semantic segmentation tasks is really expensive. And the task is relatively new compared to recent progress in ZSL for classification, detection tasks. 2. Good results have been achieved compare to other methods.

Weaknesses: - The novelty of this paper is limited and a bit incremental compared to [1]. It seems the differences are pair-wise and list-wise consistency losses which are not used in [1]. The feature aggregation step is pretty standard. The contribution of the proposed relation aggregation steps (i.e., Eqn 3) is not justified in the expeirments. A baseline to compare it to compute M^A, M^X explicitly without the relation aggregation module. - There is nothing special that is designed for segmentation tasks other than you are using a deeplab3+ as the backbone network. Thus, why not try these things on classification tasks? If the whole framework is about ZSL for segmentation, I'd like to more modules that are modeling the spatial information in images. I think this would be more important for segmentation. - What are the word embedding used in the experiments? - Detailed implementation details are missing, like learning rate, optimizer, etc. - The presentation could be further improved, eg., L140, simultaneous --> simultaneously, L175, focus --> focusing. Please carefully examine the paper.

Correctness: Seems correct.

Clarity: Could be improved. There are many grammar mistakes.

Relation to Prior Work: I would like to see more discussions with [1]

Reproducibility: No

Additional Feedback: The authors addressed some of my concerns in the rebuttal, particularly about the relations to [1] and clarifications about the feature aggregation module. The reason I'm not excited about the proposed pairwise consistency loss is that it is used a lot in zero-shot learning classification tasks. Overall, I'm satisfied with the rebuttal and I am changing my score to 6.


Review 4

Summary and Contributions: This paper proposed a Generalized Zero-Shot Segmentation method to generate the better visual features, by considering the relationship of deferent categories in both the word embedding space and the visual feature space. The experimental results show the effectiveness of the proposed method.

Strengths: (1) The proposed method integrates both feature generating and relation learning in a unified network architecture. (2) The proposed method introduces the relational constraints from different structure granularities, to facilitate the generalization of unseen categories. (3) The experimental results demonstrate the good effectiveness of the proposed method.

Weaknesses: (1) Although consider the consistence of the structural relationship of word embedding space and visual feature space is new in GZS3, it has been applied in other tasks, e,g., visual recognition. However, these references are not mentioned. (2) I suggest the author clarify the contributions of this paper. Similar works related to each contribution should be discussed. (3) Why the author do not dynamically learn the weights of the three consistency losses? (4) Why different categories are with similar relations in semantic word embedding space and visual feature space is not quite clear. Although Fig. 4 shows the relations in both spaces, I suggest the authors give more intuitive presentations. Because this is the fundament of the motivation of this paper. (5) The authors present some failure cases in the supplementary materials, but the reason is not given, for example, why the proposed method doesn’t work when with multiple instances? (6) In Fig.4, it seems that the proposed method learns a more consistent relationship. But some relation is not reasonable, i.e., motorbike and horse, which is weak in the word embedding and visual feature spaces, but it is stronger in the generated visual space with the proposed method.

Correctness: Yes

Clarity: Yes, this paper is clear to read.

Relation to Prior Work: See the weakness. Some methods have considered the consistence of the structural relationship of word embedding space and visual feature space, the author should discuss these methods. Besides, some recent GAN based zero-shot methods are missing, for example, Generative Dual Adversarial Network for Generalized Zero-shot Learning. CVPR 2019 GTNet: Generative Transfer Network for Zero-Shot Object Detection. AAAI 2020

Reproducibility: Yes

Additional Feedback:

[Author Response · NeurIPS 2020]

We appreciate the careful readings and constructive comments. We are encouraged by the reviewer's appreciation of
the proposed effective CSRL model[**R1**,**R2**,**R4**] designing for the valuable generalized zero-shot semantic segmen-
tation (GZS3) task[**R3**]. Moreover, all reviewers recognized the *superior* performance of our simple yet effective
method.[**R1**,**R2**,**R3**, **R4**]. Here we emphasize our main contributions. First, CSRL constrains the feature generation of
unseen categories by preserving relation consistency between seen and unseen categories(§4.2), which is *not* exploited
by [1]. Second, CSRL exploits the class co-existence by feature and relation aggregation(§4.1). Thus we could not only
better learn a feature generator, but also implicitly model the category coexistence in a scene (e.g. the 'cow' is usually
on the 'grass'). Such inter-class relationship is *not* explored in classification-based zero-shot task.

[**R1**,**R3**] **Discussion about the differences w.r.t [1].** Our method implicitly applies constraints to unseen categories
by exploring the relations between seen and unseen categories for the feature generation, while [1] purely employs
the seen categories to learn the feature generator, leading to a poor representative ability for the generated unseen
features. Specifically, beyond the point-wise consistency of seen classes as adopted by [1], our method further exploits
the relations between unseen and seen classes by pair-wise and list-wise consistency (§4.2). Compared with [1], our
superior performance can well demonstrate the effectiveness of relation modeling.

[**R1**] **The choice of temperature** $\gamma$**.** We discuss the effect of $\gamma$ in supplementary §C. In brief $\gamma$ is chosen by grid
searching the highest hIoU. We experimentally find that the model is robust with $\gamma$ under different unseen splits.

[**R1**] **Better to conduct extra evaluation on datasets such as ADE20k.** Completely agree. To fairly compare with
[1], we conduct experiments on object segmentation dataset (Pascal VOC) and scene parsing dataset (Pascal Context)
in this submission. However, due to the limited rebuttal period, we cannot provide the results on large=scale datasets
ADE20K and COCO. We promise to include them in the updated version.

[**R2**] **Whether the pixels of unseen classes are used.** No. We strictly follow the zero-shot settings described in §3,
thus the pixels and visual features of unseen classes are *never* used during training. In §4.1, the input nodes are the
semantic word embeddings of both seen and unseen classes. The output nodes are the generated visual embeddings. A
classifier is fine-tuned on these generated visual features. Thus, we can segment images with both seen and unseen
classes. We will further polish the descriptions in §4.1 to alleviate misunderstandings.

[**R2**, **R3**] **Discussion about the difference w.r.t other feature generation methods, e.g. zero-shot image classifica-**
**tion.** The difference is described in the second contribution given above. To further validate the argument that our
method works well for the semantic segmentation task, we run two state-of-the-art zero-shot image classification
methods ([A],[B] with publicly available code) on Pascal-VOC benchmark. Our method achieves a clear performance
boost over other classification based ones as shown in Table 1.

[A] Kampffmeyer, Michael, et al. "Rethinking knowledge graph prop-
agation for zero-shot learning." CVPR. 2019. [B] Huang, He, et al.
"Generative dual adversarial network for generalized zero-shot learning."
CVPR. 2019.

Table 1: Comparison on VOC dataset.

| Method | Seen mIou | Unseen mIoU | hIoU |
|---|---|---|---|
| DGP [A] | 72.9 | 41.7 | 53.0 |
| GDAN [B] | 73.0 | 39.8 | 51.5 |
| Ours | 73.4 | 45.7 | 56.3 |

[**R3**] **A baseline model without relation aggregation.** This baseline
(CSRL w/o relation) achieves 73.0%/43.2%/54.3% in terms of Seen
mIoU/Unseen mIoU/hIoU, which validates the effectiveness of mutual
feature and relation aggregation. Detailed results will be updated.

[**R3**] **Detailed implementations e.g. word embedding and learning rate.** The implementation details to re-produce
our results are given in §B in the supplementary material.

[**R4**] **More related works should be mentioned.** We have added the missing GAN-based methods and a thorough
related works discussion will be updated.

[**R4**] **Why not dynamically learn the weights of losses.** In this work, we focus on exploring the relation consistency
between seen and unseen categories. To maintain simplicity, we do not dynamically adjust the weights of losses. Even
this we have already reached a superior performance, and a better result could be achieved by adopting techniques in
e.g. Sener, Ozan, and Vladlen Koltun. "Multi-task learning as multi-objective optimization." NeurIPS. 2018.

[**R4**] **Intuitive discussion about the similar relations in visual and semantic.** There intrinsically exists a similar
relation among categories in both visual and semantic spaces due to the class *coexistence* and *correlation*. For example,
animals (*e.g. cat, dog*) tend to appear simultaneously or highly correlated in both visual scenes or in text corpora.

[**R4**] **The reason of failure cases.** The reason of failure cases caused by, (i) similar classes (row 1&2&4): Some unseen
classes tend to be classified as similar seen ones; (ii) highly occlusion (row 1&2): the areas which are highly occluded
by multiple instances or other objects tend to be mis-segmented; (iii) complex scene (row 3): our model fails to correctly
parse the image with a complex scene. However, for these failure cases our method is still visually better than [1].

[**R4**] **Stronger or weaker relations in Fig.4.** The consistent losses aim to constrain the relation consistency. However,
the relation cannot be exactly the same. Moreover, each row in the relation matrix is normalized in Fig.4. Thus, some
categories may be a little bit weaker or stronger compared to semantic space.

[Meta-Review · NeurIPS 2020]

Paper originally received a set of somewhat mixed reviews from four reviewers, with scores: 8, 5, 5, 6. Generally, the reviewers liked the work, commenting on how it addressed an important problem [R3] and presented a well-motivated idea [R1] that was novel [R2], simple and reproducible [R1]; ultimately resulting in good results [R1,R2,R3,R4]. Some shortcoming were also identified, including (1) unclear positioning and potential limited novelty with respect to [1] [R1,R2,R3] and (2) lack of sufficient comparisons to related work [R2,R3,R4]. Authors have provided a very through rebuttal that addressed all major concerns; providing compelling clarification of novelty (1) and additional experiments to address reviews comments for (2). As a result R2 and R3 raised their scores arriving at the final unanimously positive ratings for the paper of: 8, 7, 6, 6. AC has read the reviews, the rebuttal, resulting discussion and the paper itself. AC agrees with reviewers that the paper proposes an interesting solution for a challenging problem, with significant improvements over SoTA. Based on the consensus of reviewers and the AC, the decision is to Accept the paper.